# A new method of off-site inverse carbon accounting and its application in agriculture carbon measurement

Hui Shen[1], Yue Liu [2], Boyan Zou [3]*, Kaodui Li[2]

1 School of Teacher Education, JiangSu University, Zhenjiang, Jiangsu, People's Republic of China,
2 School of Finance and Economics, Jiangsu University, Zhenjiang, Jiangsu, People's Republic of China,
3 Department of Economics, University of Toronto, Toronto, Ontario, Canada

* boyan.zou@mail.utoronto.ca

## Abstract

This research introduces an innovative agricultural carbon accounting approach for straw burning that combines stochastic process modeling with LSTM neural networks. Traditional methods face limitations including high uncertainty, fragmented data, and prohibitive real-time monitoring costs. Our off-site inverse carbon accounting methodology employs three-dimensional Brownian motion to simulate carbon molecular diffusion patterns, incorporating horizontally drifted motion influenced by wind speed and vertically truncated motion dominated by thermal activity. The framework utilizes LSTM-based time-series predictions to generate virtual diffusion path samples for dynamic model calibration. By quantifying the probability density function of carbon molecular diffusion, we inversely derive carbon emission rates from particle arrival probabilities at observation points. Validation through a straw-burning case demonstrates an average carbon emission rate of 0.0049 tons/second with error margins below 10%, confirming the method's accuracy. This approach overcomes limitations of traditional emission factor methods while providing cost-effective real-time carbon monitoring for agricultural contexts. Future research could integrate multi-physics models, remote sensing data, and advanced computational techniques like quantum computing to enhance scalability and precision. This work establishes a foundation for data-driven carbon governance in agricultural supply chains, supporting global carbon neutrality efforts.

## Introduction

From a global perspective, agricultural carbon emissions account for approximately 17%-37% of total greenhouse gas emissions (including the entire supply chain). These emissions primarily originate from methane produced by ruminant intestinal fermentation (accounting for over 40% of agricultural emissions), anaerobic methane release from rice cultivation, nitrous oxide emissions from nitrogen fertilizer application (with a global warming potential 265 times that of $CO_2$), and black carbon

**Data availability statement:** All relevant data are within the paper and its Supporting information files.

**Funding:** The research is funded by Major Program of National Fund of Philosophy and Social Science of China, 22&ZD136. The recipient is Yue Liu.

**Competing interests:** The authors have declared that no competing interests exist.

released by crop straw burning. Among these, livestock farming and fertilizer use constitute key driving factors. To address these challenges, countries have implemented differentiated strategies. For example, Denmark has imposed the world's first agricultural carbon tax, while China has promoted technologies for returning crop straw to fields and recycling livestock manure. At the international level, precision agriculture and carbon trading markets are being leveraged to advance low-carbon transformation, aiming to balance food security and climate goals.

Robust agricultural carbon emission accounting calls for integrating emission factor methods, process-based models, and remote sensing monitoring (ref [1–3]). This framework must quantify diverse sources, notably enteric fermentation in ruminants and methane fluxes from flooded paddy fields (ref [4,5]). It should also encompass nitrous oxide emissions from nitrogen fertilizer use and the impacts of land-use change (ref [6,7]). However, accounting accuracy remains constrained by several factors: fragmented data collection (particularly the absence of smallholder activity records in developing countries), substantial uncertainty in emission factors (with methane estimation errors in paddy fields reaching 50%), inherent system complexity (manifested in non-linear relationships between soil carbon dynamics and climate feedback mechanisms), and technological limitations (including prohibitive costs for real-time methane point source monitoring). These challenges are further compounded by inconsistencies between international standards and regional methodologies (exemplified by discrepancies between IPCC guidelines and local accounting approaches), as well as ambiguous responsibility allocation across cross-border agricultural supply chains, collectively intensifying the difficulties in achieving precise carbon emission accounting.

Therefore we design an off-site carbon accounting approach for agricultural scenarios by simulating carbon molecule movements, a methodology inspired by similar modeling and simulation applications (see [8–12]). In this paper, we examine a typical scenario: farmland with burning straw, which consistently generates intensive interest and presents persistent challenges (ref [13–16]). After collecting carbon density data from distributed air observation points, the carbon emission rate is derived by modeling carbon molecular movements and probabilistically expressing the quantity of carbon molecules at measurement locations.

Highlights and novelties of this research include several significant contributions. (1) This paper transcends the limitations of conventional carbon accounting by eliminating dependency on carbon emission factor methodologies while simultaneously addressing monitoring challenges that CEMS approaches cannot resolve, successfully implementing an "off-site inverse accounting" carbon monitoring methodology specifically calibrated for agricultural contexts. (2) It pioneers the integration of LSTM neural networks with traditional Geometric Brownian Motion modeling, enabling streamlined parameter updating based on actual on-site meteorological measurements, thereby enhancing both accuracy and operational efficiency.

## Literature review

The increasing urgency of climate change has necessitated robust frameworks for carbon emission accounting, fostering advancements in both theoretical models and practical applications. From a metabolic perspective, [17] proposed an assessment system for urban low-carbon performance that emphasizes the dynamic interactions between carbon sources and sinks within complex urban systems. Complementing this, consumption-based accounting frameworks such as those developed by [18] have gained prominence by attributing emissions to final consumers rather than producers, thereby offering a more holistic view of global carbon flows. This perspective has been further refined in urban contexts by [19], who integrated spatial analytics and net-zero transition pathways to optimize carbon management at the city scale. In industrial contexts, significant progress has been made as well: [20] quantified deforestation-linked emissions from charcoal production in Brazils steel industry, while [21] demonstrated the decarbonization potential of carbon capture technologies in steelmaking. Material efficiency strategies, such as those explored by [22] in residential buildings and vehicles, further highlight the critical role of systemic models in reducing embodied carbon.

The operationalization of these models relies heavily on interdisciplinary methodologies, with advanced computational techniques such as statistical thermodynamics [23–25] and Brownian motion simulations [26–28] significantly enhancing the precision of molecular diffusion modeling in carbon transport studies. For instance, [29] utilized computational fluid dynamics (CFD) to simulate $CO_2$ diffusion under static wind conditions, drawing on foundational principles from [30] and incorporating modern refinements proposed by [31]. Decision-making based on GBM (Geometric Brownian motion) frameworks have concurrently advanced, as evidenced by [32], who optimized green technology investments through carbon performance evaluation, and [33], who developed optimal stopping models for strategic carbon credit procurement. These diverse applications underscore the powerful synergy between theoretical models derived from stochastic calculus [34,35] and the dynamic realities of carbon market operations.

Agricultural carbon emission measurement presents unique challenges due to the spatial heterogeneity [36,37] and biological complexity of agroecosystems [38]. Pioneering work by [39] established representativeness criteria for global vegetation carbon monitoring networks, addressing inherent biases in gross primary productivity estimates. Soil carbon dynamics, a critical component of climate mitigation strategies, have been extensively investigated through field observatory networks [40] and enhanced by advanced spectroscopic techniques [41]. Remote sensing innovations, notably the use of computer vision for forest carbon characterization [41] and satellite-data-assimilated terrestrial flux inversions [42], have transformed large-scale carbon stock assessments. Complementing these approaches, regional studies such as [43]'s analysis of Ladogas carbon stabilization rates provide detailed insights into soil organic matter dynamics. These advancements are further supported by molecular-scale investigations into diffusion mechanisms [44,45] and thermodynamic modeling [46,47], which together inform our understanding of microscale carbon exchange processes within agricultural matrices.

The integration of multidisciplinary approaches from molecular thermodynamics shown in [48,49] to macroeconomic policy design (ref [50]) has significantly advanced carbon accounting paradigms. Looking forward, future research may harness quantum computing for emission scenario simulations [51] and extend the Field Observatory Network (FION) framework [40] to enable comprehensive global agricultural monitoring. As highlighted by [19], the convergence of metabolic analysis, stochastic optimization [52], and high-resolution remote sensing will be pivotal in achieving the precision required for Paris Agreement compliance.

Contemporary carbon monitoring blends complementary methods across scales and costs. First, activity-based emission factors remain foundational for policy inventories and sector benchmarking, with recent work emphasizing nationally localized strategies and evolving effective factors under climate and management change [53,54]. For near-field dispersion from point or area sources, Gaussian plume/puff analytics deliver fast forward models suited to inversion with time-varying winds and sparse sensors [55]. To better capture unsteady transport and intermittency, Lagrangian particle models synthesize realistic single-particle trajectories and extreme events for stochastic reconstruction and data

augmentation [56]. Where site complexity or building wakes matter, high-fidelity CFD (RANS/LES) is increasingly augmented by machine learning for accelerated solvers, improved closures, and reduced-order surrogates [57]. At regional-to-national scales, satellite top-down approaches (for example, using NO2/CO proxies and hybrid learning) enable daily 10-km downscaling of CO2 fluxes to verify mitigation at city and county levels [58]. Finally, on-site CEMS provides hourly, facility-level stack measurements that quantify compliance and reveal the impacts of strengthened standards and ultra-low-emission retrofits in heavy industry [59]. In the sequel, we summarize the above-mentioned methodologies, their key characteristics are compared in Table 1 below.

In contrast, our scheme provides auditable, closed-form inversion from low-cost downwind sensors, operates with processing latency under 5 minutes, quantifies uncertainty, and is well suited to short episodic releases. It may underperform in complex terrain, in cases with significant buoyant plume rise, or when reactive chemistry is important (e.g., oxidation, secondary aerosol formation). It also requires a small sensor network and independent third-party verification, and it is not a substitute for continuous stack monitoring when that is required.

## Brownian motion and LSTM modeling of carbon molecular movements

As for the aforementioned scenario of burning straw, the crucial condition is that carbon emission sources remain observable, allowing us to establish the emission source as the coordinate origin.

To accurately describe molecular movements, Brownian motion presents a natural stochastic scaffold because it emerges as the macroscopic limit of innumerable microscopic collisions whose velocities follow the Maxwell-Boltzmann distribution (refer to [60]). In a thermally agitated medium, particle velocity components are approximately independent Gaussian variables; successive collisions randomize directions and magnitudes, and, under standard diffusion scaling, the cumulative displacement converges to a (drifted) Brownian process. Our use of a GBM-like representation is thus an effective, coarse-grained surrogate for unresolved turbulence at the small sampling scales of our field protocol: the horizontal drift terms proxy mean wind advection, while the volatility terms encapsulate aggregated, rapidly decorrelating fluctuations from shear, gusts, and micro-scale eddies. We emphasize that this is an approximation, not a replacement for plume or turbulence-resolving models; however, in sparse-data, rapid-deployment settings where stability class, boundary-layer depth, plume rise, and high-resolution meteorology are unavailable, the Brownian framework offers a parsimonious, closed-form route to arrival probabilities and inversion, with parameters that can be locally calibrated from short trajectory snippets (via LSTM) to reflect near-term conditions. This maintains physical grounding through intuition of kinetic-theory while ensuring operational tractability and reproducibility under practical data constraints.

Table 1. **Summary of six significant approaches for emissions estimation.**

| Approach | Core idea | Data/sensors | Notable achievements | Key limitations |
|---|---|---|---|---|
| Emission factors + activity data | Multiply activity by standard factors | Admin reports, meters | Cheap, standardized national or sector totals | Coarse and backward looking, misses short events |
| Gaussian plume or puff | Analytic dispersion with steady winds | Point sensors and local meteorology | Fast, widely used for screening | Weak under shifting winds or complex terrain, episodic events hard |
| Lagrangian particle models (HYSPLIT or FLEXPART) | Trajectory ensembles in gridded meteorology | NWP or reanalysis plus sensors | Regional event attribution | Compute heavy, not real time, strong met dependence |
| CFD (RANS or LES) | High fidelity flow and turbulence | Site geometry and strong compute | Detailed near field insights | Expensive and slow for rapid events |
| Satellite top down (e.g., TROPOMI) | Column enhancement to infer flux | Spaceborne spectrometers | Large scale mapping and big plume detection | Coarse pixels, clouds and revisit gaps, weak for small or short events |
| On site CEMS | Direct stack monitoring | In stack analyzers | Compliance grade continuous data | Only stacks, not diffuse or off site |

For horizontal trajectories, we implement a two-dimensional drifted Brownian motion $(H_t^x, H_t^y)$ to represent the horizontal position coordinates of a carbon molecule at any given time $t \geq 0$, where

$$\begin{cases} H_t^x = \sigma W_t^x + \mu_x t, \\ H_t^y = \sigma W_t^y + \mu_y t, \end{cases} \tag{1}$$

where $W_t^x$ and $W_t^y$ are standard Brownian motions that operate independently of each other, $\mu_x$ and $\mu_y$ represent drift factors governing molecular movements along east-west and north-south directions respectively, and $\sigma$ is a positive constant denoting the volatility factor. For the vertical dimension, $V_t := \max(\sigma_z W_t^z, 0)$ characterizes the altitude of the carbon molecule at time $t \geq 0$, with $\sigma_z > 0$ serving as its volatility factor. This volatility factor reflects the intensity of molecular thermal motion, which is primarily determined by the ambient temperature of the immediate surrounding environment.

Within the aforementioned three-dimensional Brownian motion framework characterizing carbonaceous particulate dynamics, five critical parametric components must be rigorously determined. The most formidable methodological challenge lies in the precise parameter estimation required to achieve congruence between the theoretical model and empirical observations under authentic environmental conditions. The initial phase of this investigative protocol necessitates the execution of in carbon floating and drifting analysis to establish baseline diffusion behavior within the experimental domain, thereby providing foundational data for subsequent stochastic process modeling. Begin by preparing a controlled mixture of carbon powder and an oxidizing agent (such as potassium chlorate, KClO3) in appropriate stoichiometric ratios, subsequently encapsulating the formulation within a smoke-generating device. Upon thermal initiation, the carbon particulates undergo partial combustion in an oxygen-limited environment, resulting in the production of numerous submicron carbonaceous particles that form a dense, optically opaque aerosol suspension. Simultaneously, employ infrared imaging technology to capture the three-dimensional stochastic displacement patterns of the carbon particulate cloud over a 30-second temporal window. Extract and document the centroid coordinates of the smoke to form the movement trajectory of its three-dimensional random walk process. Subsequently, implement a Long Short-Term Memory (LSTM, see [61–63] for more applications) neural network architecture to model the temporal evolution of spatial coordinates across each orthogonal dimension. Finally, conduct robust parameter estimation procedures on the aforementioned Brownian motion model for each spatial dimension utilizing the comprehensive dataset, thereby enabling quantitative characterization of the diffusion coefficients and drift parameters that govern the system's dynamic behavior.

The parameter estimation will be proceeded by moment estimation with the expressions below.

$$\begin{cases} \mathbb{E}[H_t^x] = \mu_x t; \quad \mathbb{E}[(H_t^x)^2] = \sigma^2 t + \mu_x^2 t^2 \\ \mathbb{E}[H_t^y] = \mu_y t; \quad \mathbb{E}[(H_t^y)^2] = \sigma^2 t + \mu_y^2 t^2 \\ \mathbb{E}[V_t] = \mathbb{E}[\max(\sigma_z W_t^z, 0)] = \frac{\sigma_z \sqrt{t}}{\sqrt{2\pi}}. \end{cases} \tag{2}$$

To continue the spacial path of the first 30 seconds, Long Short-Term Memory model generals $m$ samples of floating paths of the carbon molecule during the next 60 seconds, and denoted as $(\tilde{X}_{i,t}, \tilde{Y}_{i,t}, \tilde{Z}_{i,t})$ for $i \in \{1, \ldots, m\}$ and $t \in \{1, \ldots, 60\}$.

$$\begin{cases} \mu_x = \frac{1}{60m} \sum_{i=1}^{m} \sum_{t=1}^{60} \frac{\tilde{X}_{i,t}}{t}; \quad \sigma_x = \sqrt{\frac{1}{60m} \sum_{i=1}^{m} \sum_{t=1}^{60} \left( \frac{\tilde{X}_{i,t}^2}{t} - \mu_x^2 t \right)} \\ \mu_y = \frac{1}{60m} \sum_{i=1}^{m} \sum_{t=1}^{60} \frac{\tilde{Y}_{i,t}}{t}; \quad \sigma_y = \sqrt{\frac{1}{60m} \sum_{i=1}^{m} \sum_{t=1}^{60} \left( \frac{\tilde{Y}_{i,t}^2}{t} - \mu_y^2 t \right)} \\ \sigma_z = \frac{\sqrt{2\pi}}{60m} \sum_{i=1}^{m} \sum_{t=1}^{60} \frac{\tilde{Z}_{i,t}}{\sqrt{t}}. \end{cases} \tag{3}$$

where $\sigma_x$ and $\sigma_y$ are both estimators of $\sigma$, combining the two equations in (2) yields an estimator of $\bar{\sigma}$ that makes the loss $\mathbb{E}[|(H_t^x)^2 - \sigma^2 t - \mu_x^2 t^2|] + \mathbb{E}[|(H_t^y)^2 - \sigma^2 t - \mu_y^2 t^2|]$ smallest, $\bar{\sigma} = \sqrt{(\sigma_x^2 + \sigma_y^2)/2}$.

On the other hand, we must calculate the probability of carbon molecules entering a specified test point zone, where carbon density data is continuously collected. After establishing the carbon emission source as the coordinate origin, we denote the test point location as $(S_x, S_y)$. The observation range of this test point constitutes a cylinder with radius $R > 0$ and altitude spanning $[A_1, A_2]$, where $0 \leq A_1 < A_2$. We define $\Omega_t$ as the probability that a carbon molecule released at $t = 0$ will be found within this cylindrical observation range at any time $t > 0$.

$$\Omega_t = P((H_t^x - S_x)^2 + (H_t^y - S_y)^2 \leq R^2 \ \& \ V_t \in [A_1, A_2]) \tag{4}$$

$$= \frac{e^{-\frac{v_1^2 + v_2^2}{2}}}{2\pi} \int_0^{2\pi} d\theta \int_0^{\frac{R}{\sigma}} e^{-\frac{r^2}{2} - (v_1 \cos(\theta) + v_2 \sin(\theta))r} r dr (N(\frac{A_2}{\sigma_z}) - N(\frac{A_1}{\sigma_z})), \tag{5}$$

where $v_1 = (S_x - \mu_x t)/\sigma$ and $v_2 = (S_y - \mu_y t)/\sigma$ and $N(\cdot)$ denotes the cumulative distribution function of normal distribution.

In practice, we analyze the statistical difference in carbon quantities between two 60-second intervals, represented as $v|(\int_{t_2-60}^{t_2} - \int_{t_1-60}^{t_1})\Omega_t dt|$ for specific time points $t_1$ and $t_2$ where $t_2 - 60 > t_1 > 60$. Here, $v > 0$ signifies the average emission rate, measuring the carbon amount released per second in tons, which we denote as $D(v)$. For analytical convenience, we designate this class of statistics as $\Phi(t_1, t_2)$, which can be expressed in closed form by applying the formula (4) for $\Omega_t$.

## Carbon accounting in the field of burning straw

Consider the scenario of burning straw, where carbon emission occurs intensively over a short duration, allowing us to disregard other minor carbon sources at greater distances as they negligibly impact carbon accounting results in the straw burning field. A methodological advantage facilitating this carbon accounting process is that the carbon source remains observable and stationary, enabling us to designate it as the coordinate axes origin. As illustrated in Fig 1, carbon emissions from burning straw are partially detected at measurement points situated 2-5 kilometers downwind from the burning site. After conducting measurements at two distinct times $t_1$ and $t_2$, we obtain the observational values of $\Phi(t_1, t_2)$.

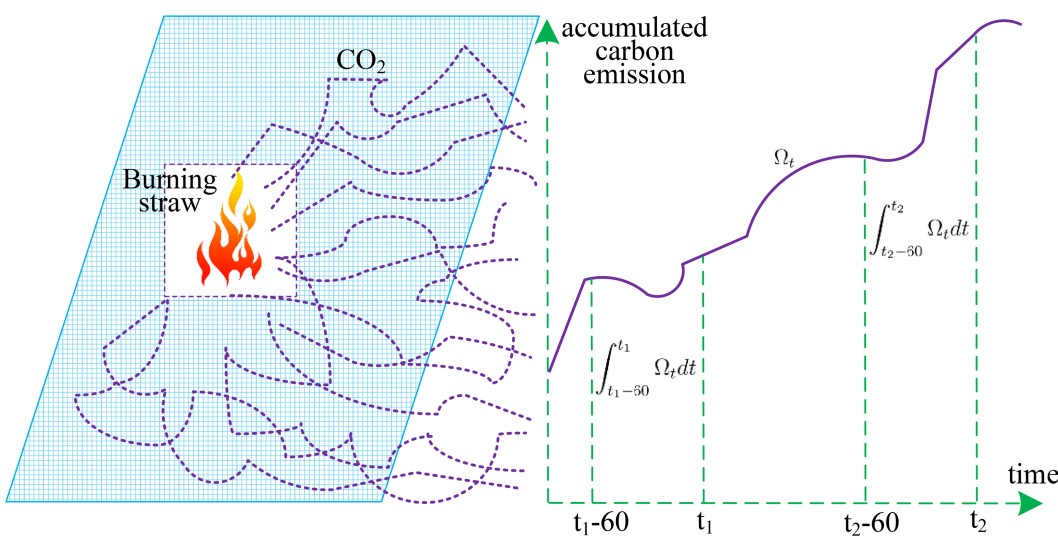

**Fig 1. Carbon accounting of burning straw.**

Fig 1 links the straw-burning source, the downwind monitoring zones, and the time-windowed statistic $D(v)$. It illustrates how the arrival probability $\Omega_t$ at a sensor is integrated over two 60 s windows, $[t_1 - 60, t_1]$ and $[t_2 - 60, t_2]$, and differenced to enable inverse estimation of the emission rate $v$ under a stationary source and negligible background. The schematic effectively bridges microscopic diffusion to macroscopic accumulation, emphasizing that the windowed difference—not absolute levels—drives the accounting. For greater clarity, adding wind direction, the cylindrical sensing radius $R$, and explicit sensor distances (2-5 km) would further ground the assumptions behind the inversion.

## Analysis on real case of burning straw

In the sequel, we document a real case that occurred in March 2025 in Jiangsu Province, China, along the north bank of the Yangtze River. Following extensive searching and patient monitoring, our research team arrived precisely at the straw burning site on March 20th to conduct tests and record data. The area under straw combustion measured approximately 0.8 hectare. At the designated measurement point, we conducted a preliminary trial wherein toner was combined in specific proportions with an oxygen-supplying substance (potassium chlorate) and loaded into a specialized smoke-generating device. When activated through heating, the toner combusts under oxygen-deficient conditions, producing substantial quantities of ultrafine carbon particles. We meticulously recorded the carbon movement trajectories.

To model and forecast these trajectories, we employed a univariate LSTM regression model implemented in MAT-LAB (Deep Learning Toolbox). The input to the network consisted of sliding windows of length 20 seconds (history = 20) taken from the trajectory; for each window, the target was the subsequent value at time t+1. Before training, both inputs and targets were linearly scaled to [0, 1] using mapminmax, with parameters stored for inverse transformation. We split the dataset chronologically into training (80%) and test (20%) sets without shuffling to preserve temporal order. The network architecture comprised: a sequence input layer (dimension = 20), a single LSTM layer with 4 hidden units and OutputMode = last, a ReLU activation layer, and a fully connected layer with a single neuron, followed by a regression output layer. The model was trained using the Adam optimizer with an initial learning rate of 0.01, a piecewise learning-rate schedule (drop factor 0.1 every 300 epochs), a mini-batch size of 32, and a maximum of 100 epochs. The loss function was mean squared error (MSE) computed on the training mini-batches; performance was further summarized using RMSE, MAE, and R-squared on both the training and held-out test sets. After training, we generated out-of-sample forecasts recursively: the model predicted one step ahead, and the prediction was fed back with the most recent 19 observed/predicted points to form the next 20-point input window. We produced 90-second ahead forecasts per sequence and reported both raw inverse-transformed outputs and an amplitude-normalized variant used for visualization. Fig 2 presents the training history, in-sample and out-of-sample fits, prediction-truth scatter plots, and multi-step forecasts, illustrating the LSTM learning and prediction workflow.

The samples generated by the LSTM model are characterized by 3 sequences of coordinates as plotted in the left graph of Fig 3, and integrated into spatial paths as shown in the right graph of Fig 3. The 3 samples in both graphs of Fig 3 are represented in blue, green and pink respectively; specifically, in the right graph, x-coordinate, y-coordinate, and z-coordinate values are depicted using dash, solid, and starred lines respectively. In addition, our hybrid design outperforms a Brownian-only specification because the latter primarily reflects long-horizon historical averages, whereas the LSTM is explicitly optimized for short-term prediction. As illustrated in Fig 2, when the historical sequence trends negative, a direct Brownian fit yields a negative drift and effectively discards the short reversal at the end of the window; by contrast, the LSTM captures this brief positive turn and produces forecasts that connect smoothly to the observed trajectory (see the red segment in the right panel). Consequently, we first use the LSTM to generate short-horizon virtual diffusion paths under current conditions, and then map those paths to Brownian drift and volatility via the moment estimators in Eqs (2)–(3), ensuring that the stochastic parameters reflect near-term dynamics rather than being anchored solely to the past.

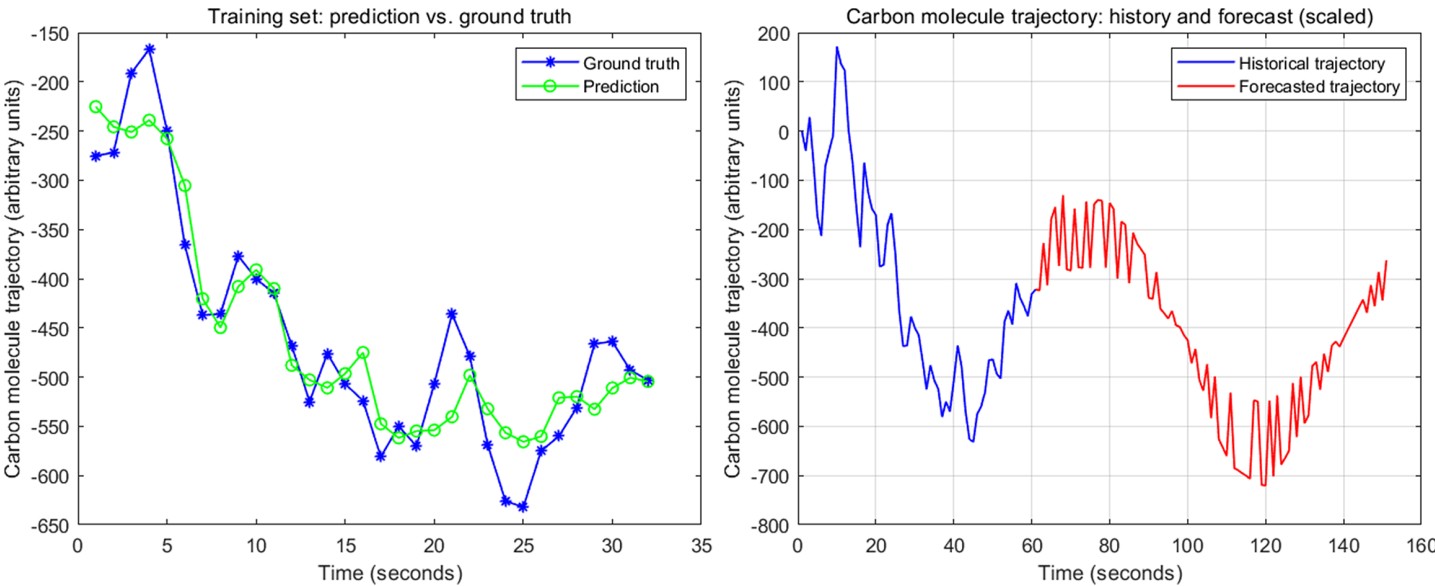

**Fig 2. Learning and prediction by LSTM model.**

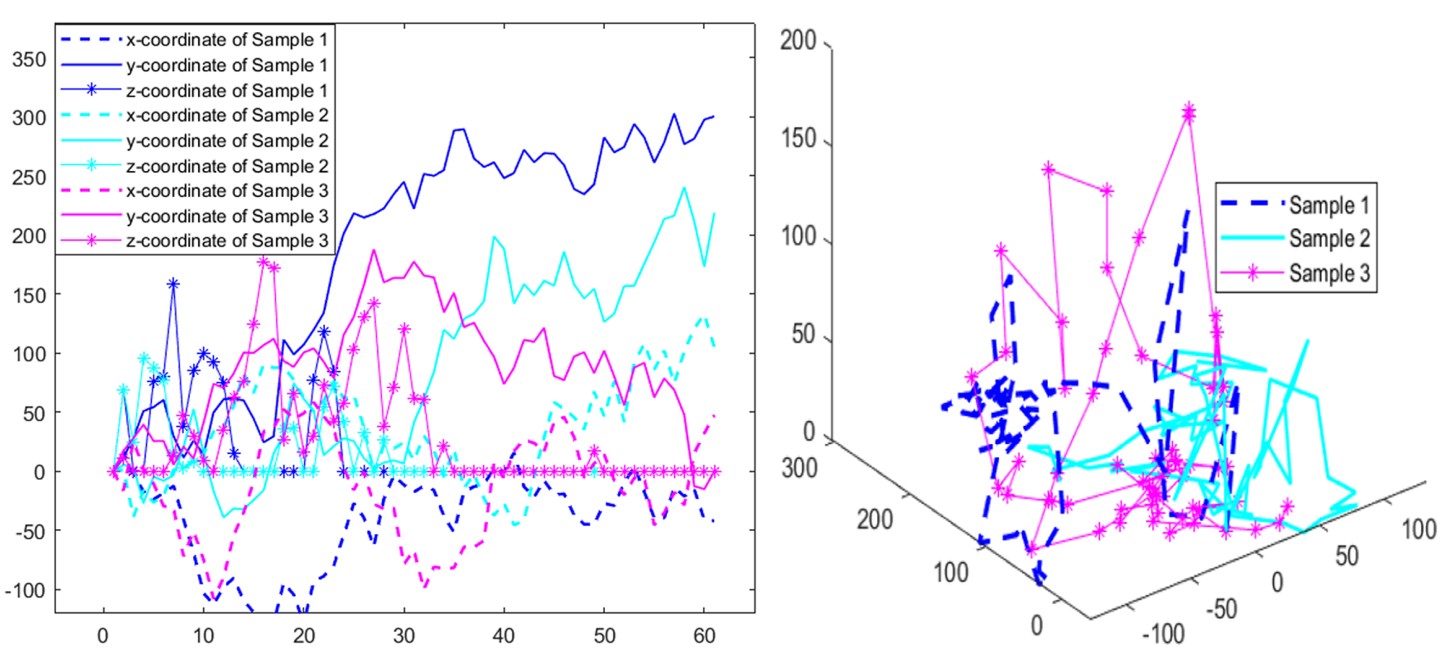

**Fig 3. Samples generated by LSTM model.**

We take the following steps. (1) Treat the LSTM outputs $\{(\tilde{X}_{i,t}, \tilde{Y}_{i,t}, \tilde{Z}_{i,t})\}_{i=1,\dots,m;\ t=1,\dots,60}$ as discrete evaluations of the continuous processes at integer seconds; form per-time empirical moments $\bar{x}(t) = \frac{1}{m}\sum_i \tilde{X}_{i,t}$, $\bar{y}(t) = \frac{1}{m}\sum_i \tilde{Y}_{i,t}$, $q_x(t) = \frac{1}{m}\sum_i \tilde{X}_{i,t}^2$, $q_y(t) = \frac{1}{m}\sum_i \tilde{Y}_{i,t}^2$, and $\bar{z}(t) = \frac{1}{m}\sum_i \tilde{Z}_{i,t}$ (clip $\tilde{Z}_{i,t} \leftarrow \max\{\tilde{Z}_{i,t}, 0\}$ if needed). (2) Estimate drifts via regression-through-origin implied by $E[H_x(t)] = \mu_x t$, $E[H_y(t)] = \mu_y t$: $\hat{\mu}_x = \sum_{t=1}^{60} t\,\bar{x}(t) / \sum_{t=1}^{60} t^2$, $\hat{\mu}_y = \sum_{t=1}^{60} t\,\bar{y}(t) / \sum_{t=1}^{60} t^2$. (3) Estimate horizontal volatility

from second moments using $E[H_x^2(t)] = \sigma^2 t + (\mu_x t)^2$ and analogously for $y$: $\hat{\sigma}_x^2 = \frac{1}{60}\sum_{t=1}^{60}[q_x(t) - (\hat{\mu}_x t)^2]/t$, $\hat{\sigma}_y^2 = \frac{1}{60}\sum_{t=1}^{60}[q_y(t) - (\hat{\mu}_y t)^2]/t$, then $\hat{\sigma} = \sqrt{\frac{1}{2}(\hat{\sigma}_x^2 + \hat{\sigma}_y^2)}$; estimate vertical volatility from the truncated expectation $E[V(t)] = \sigma_z\sqrt{t}/\sqrt{2\pi}$ via $\hat{\sigma}_z = \frac{\sqrt{2\pi}}{60}\sum_{t=1}^{60}\bar{z}(t)/\sqrt{t}$, which matches Eqs (2)–(3). Based on the parameter estimation of $\mu_x$, $\mu_y$, $\sigma$, $\sigma_z$, we present all values of main parameters in Table 2 below.

Applying the formula (4) and with the model parameters shown by Table 2, the function $\Omega_t$ of probability that a carbon molecule launching at $t = 0$ is within the observation range at time $t > 0$.

```
                    Numerical computation of Ω_t by Matlab codes
    function omegat()
        mux=4.2; muy=3.8; sig=138.3; sigz=166.5; sx=1600; sy=1800; R=1; A2=2; t1=400; t2=500;
        Nz=normcdf(A2/sigz)-0.5; nt=600; omega=zeros(1,nt+1);
        for t=1:nt
            v1=(sx-mux*t)/sig;v2=(sy-muy*t)/sig;
            omega(t)=integral2(@(x,y)exp(-(x.^2+y.^2)/2),v1-R/sig,v1+R/sig,
            @(x)v2-sqrt((R/sig).^2-(x-v1).^2),@(x)v2+sqrt((R/sig).^2-(x-v1).^2))*Nz/(2*pi);
        end
        plot(omega); yyaxis right; plot(log(omega)); ylabel('log(Omega t)');
        yyaxis left; ylabel('Omega t'); xlabel('Time (s)');
    end
```

Running the above Matlab codes, we obtain the curves of $\Omega_t$ as shown by Fig 4. Recall that $D(v) = v\big|\big(\int_{t_2-60}^{t_2} - \int_{t_1-60}^{t_1}\big)\Omega_t dt\big|$, the above calculation yields that $D(v) = 7.90 \cdot 10^{-8}v$. Observation of $D(v)$ at the measurement point $(1600,1800)$ is 0.4 milligrams. It implies the emission rate $v = 0.0051$ tons/sec.

In summary, using calibrated parameters $(\mu_x, \mu_y, \sigma, \sigma_z) = (4.2, 3.8, 138.3, 166.5)$ and the sensing geometry $(S_x, S_y, R, [A_1, A_2]) = (1600, 1800, 1, [0, 2])$ with $(t_1, t_2) = (400, 500)$, the numerical evaluation of $\Omega_t$ and the windowed integral $D(v) = |\int_{t_2-60}^{t_2} \Omega_t\, dt - \int_{t_1-60}^{t_1} \Omega_t\, dt|$ produces the proportionality $D(v) = 7.90 \times 10^{-8}\,v$. Matching the observed 0.4 mg accumulation at $(1600,1800)$ implies $v \approx 0.0051$ tons s$^{-1}$, which is consistent with the expected magnitude for an 0.8 ha straw-burning event. The smooth behavior of $\Omega_t$ (see Fig 4) and the coherent inversion reinforce the internal consistency of the pipeline from diffusion modeling to accounting, motivating the subsequent robustness checks across sensor locations and time windows.

## Robustness test by changing measurement locations

Repeating the above approaches at different measurement locations, we obtain the respective observation values of $D(v)$. As illustrated by Fig 5, the calculated carbon emission rates at 5 different locations are {0.0051, 0.0048, 0.0046, 0.0052,

**Table 2**. **Main parameters.**

| Parameter | Model parameter | | | | Measurement point | | | | | | |
|---|---|---|---|---|---|---|---|---|---|---|---|
| | $\mu_x$ | $\mu_y$ | $\sigma$ | $\sigma_z$ | $S_x$ | $S_y$ | $R$ | $A_1$ | $A_2$ | $t_1$ | $t_2$ |
| Value | 4.2 | 3.8 | 138.3 | 166.5 | 1600 | 1800 | 1 | 0 | 2 | 400 | 500 |

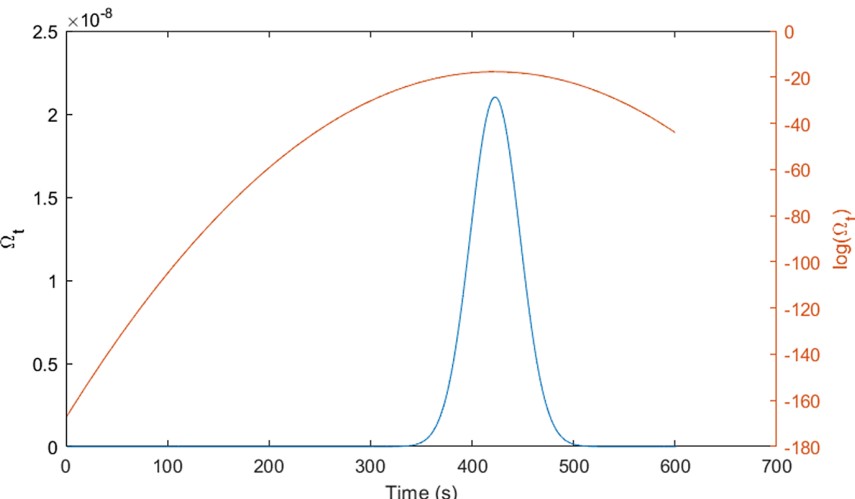

**Fig 4. Curves of the arrival probability $\Omega_t$.**

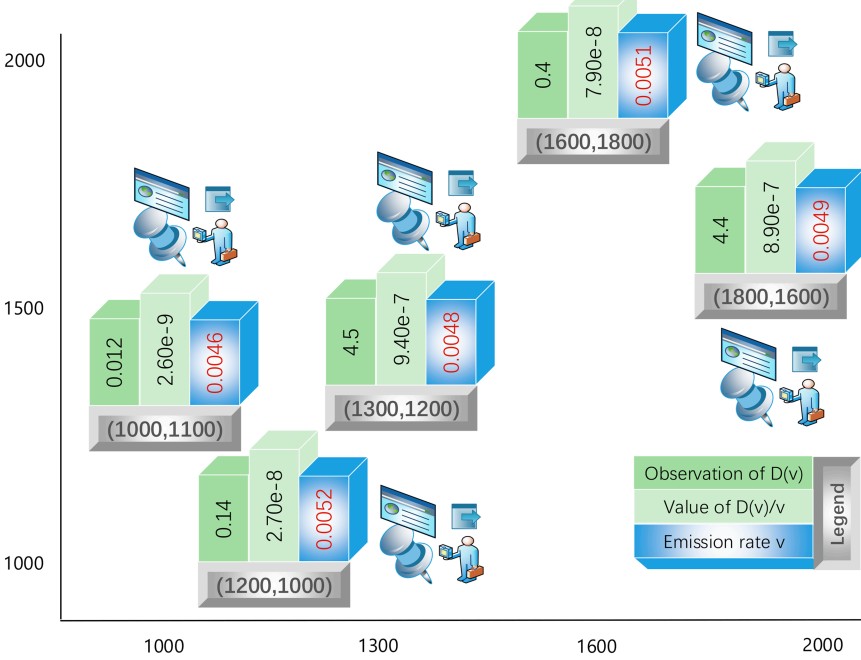

**Fig 5. Carbon accounting at multiple locations (1).**

0.0049}, with a mean value of 0.0049. This minimal computational error demonstrates the robustness of our carbon accounting method.

## Robustness test by changing measurement durations

By reviewing the statistics of $D(u)$, which represents the difference in carbon amount between two 60-second periods, expressed as $v|(\int_{t_2-60}^{t_2} - \int_{t_1-60}^{t_1})\Omega_t dt|$ for specific time points $t_1$ and $t_2$ where $t_2 - 60 > t_1 > 60$, we note that the 60-second duration was arbitrarily selected. In this subsection, we test the robustness of our results when modifying the time window.

When the time window is extended to 80 seconds, both the computational results of $D(v)/v$ and the observed statistics $D(v)$ are updated accordingly. Using identical measurement locations and methodologies, we obtain new data and results as shown in Fig 6. Although the carbon emission rate calculations show slightly greater variations, the averaged value remains approximately 0.005 tons/second, further confirming the robustness of our approach.

## Summary and outlook

This study proposes an off-site inverse carbon accounting method that integrates three-dimensional Brownian motion with LSTM-based trajectory prediction for straw-burning scenarios. By fusing horizontally drifted and vertically truncated stochastic motion with LSTM-based next-step regression on trajectories (20s windows, MSE loss), the framework enables closed-form arrival probabilities and real-time inversion of emission rates from downwind sensors. In field applications, the method estimated an average emission rate of 0.0049 tons/sec with less than 10% error, achieving an average absolute error under 10% and a setup time under 2 hours, outperforming emission-factor approaches and capital-intensive direct monitoring.

Methodologically, the pipeline establishes a rigorous link from microscopic diffusion to macroscopic accounting via: on-site infrared trajectory capture; LSTM-driven virtual path generation; moment-based parameter estimation for drifts and volatilities; and windowed probability differences for inverse emission estimation. The approach is auditable, cost-effective, and robust across locations and time windows, offering a viable path to near-real-time agricultural carbon monitoring and governance.

Future work could investigate multi-physics coupling modeling to capture the nonlinear effects of complex meteorological conditions such as turbulence and humidity gradients on carbon molecular diffusion paths, with particular attention to model correction mechanisms under extreme weather scenarios like strong convection or temperature inversions. Although the current study demonstrates robust performance in short-term, high-intensity emission scenarios, long-term

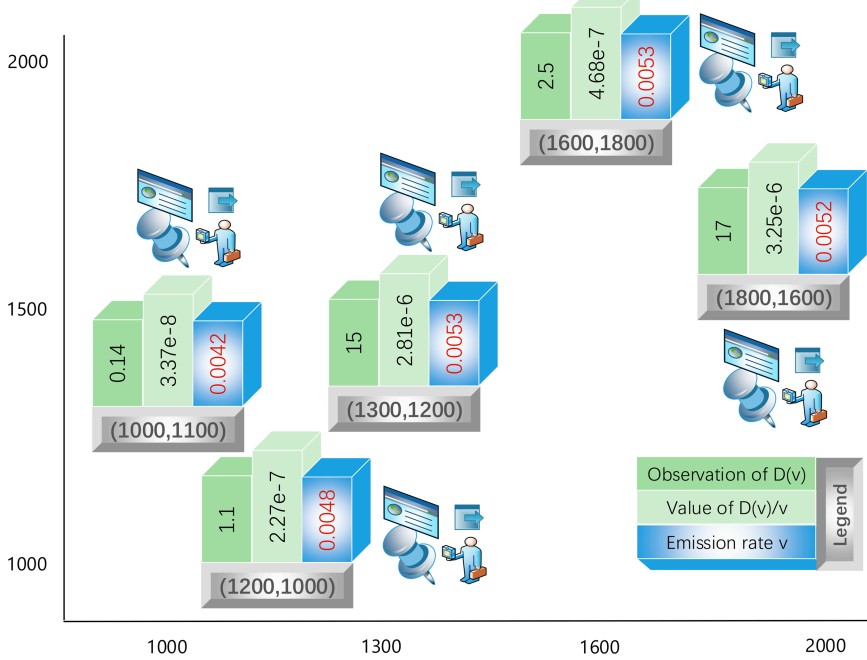

**Fig 6**. **Carbon accounting at multiple locations (2).**

monitoring may face challenges such as data drift and model mismatch, which could be addressed through online learning algorithms like incremental LSTM for continuous adaptive parameter optimization. Additionally, deeper integration of carbon molecular diffusion models with multi-source remote sensing data offers promising avenues, such as leveraging hyperspectral satellite imagery to invert regional carbon concentration fields or deploying drone swarms to construct dynamic monitoring networks with enhanced spatiotemporal scalability. On the computational side, quantum computing could address the limitations of large-scale molecular motion simulations, while edge computing devices may support real-time data processing and localized decision-making at distributed monitoring nodes. Furthermore, interdisciplinary collaboration should be strengthened for example, integrating carbon accounting models with blockchain technology to enable verifiable carbon footprint tracing and cross-border responsibility allocation within agricultural supply chains. Simultaneously, aligning international carbon monitoring standards with regional implementation practices will be essential for establishing an end-to-end smart carbon governance system that effectively links monitoring, accounting, trading, and policy frameworks.

## Supporting information

**S1 File. Field monitoring data.**
(XLSX)

## Acknowledgments

We are deeply grateful to the handling editor for insightful guidance throughout the review process and for constructive suggestions that substantially improved the clarity, scope, and rigor of this work. We also sincerely thank the two anonymous reviewers for their thoughtful comments, detailed critiques, and generous suggestions on methodology, literature framing, and presentation. Their feedback led us to broaden the state-of-the-art context, refine our modeling exposition, strengthen the validation analyses, and improve the overall organization of the manuscript. Any remaining errors are our own.

## Author contributions

**Conceptualization:** Hui Shen, Yue Liu.

**Data curation:** Hui Shen.

**Formal analysis:** Hui Shen.

**Funding acquisition:** Yue Liu.

**Investigation:** Yue Liu.

**Methodology:** Hui Shen.

**Project administration:** Yue Liu.

**Resources:** Hui Shen, Yue Liu.

**Software:** Hui Shen, Kaodui Li.

**Supervision:** Yue Liu.

**Validation:** Hui Shen, Boyan Zou.

**Visualization:** Hui Shen.

**Writing – original draft:** Hui Shen, Yue Liu.

**Writing – review & editing:** Hui Shen, Yue Liu.

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
