## [Decision Letter · Decision Letter 0]

5 Nov 2025

PONE-D-25-52174A new method of off-site inverse carbon accounting and its application in agriculture carbon measurementPLOS ONE

Dear Dr. Zou,

Thank you for submitting your manuscript to PLOS ONE. After careful consideration, we feel that it has merit but does not fully meet PLOS ONE’s publication criteria as it currently stands. Therefore, we invite you to submit a revised version of the manuscript that addresses the points raised during the review process.

We look forward to receiving your revised manuscript.

Kind regards,

Lei Zhang, PhD

Academic Editor

PLOS ONE

Journal Requirements:

2. Please note that PLOS One has specific guidelines on code sharing for submissions in which author-generated code underpins the findings in the manuscript. In these cases, we expect all author-generated code to be made available without restrictions upon publication of the work.

Please review our guidelines at https://journals.plos.org/plosone/s/materials-and-software-sharing#loc-sharing-code and ensure that your code is shared in a way that follows best practice and facilitates reproducibility and reuse.

4. Please note that funding information should not appear in the Acknowledgments section or other areas of your manuscript. We will only publish funding information present in the Funding Statement section of the online submission form. Please remove any funding-related text from the manuscript.

5. You have indicated that data is available from boyan.zou@mail.utoronto.ca. Please can we ask you to provide us with a general contact email address for the data requests, so readers can request access in perpetuity. If a general email is not available please provide a link to a website where readers can obtain access to data.

6. Please ensure that you refer to Figures 1 to 6 in your text as, if accepted, production will need this reference to link the reader to the figure.

7. Please remove your figures from within your manuscript file, leaving only the individual TIFF/EPS image files, uploaded separately. These will be automatically included in the reviewers’ PDF.

8. We notice that your supplementary figures are uploaded with the file type 'Figure'. Please amend the file type to 'Supporting Information'. Please ensure that each Supporting Information file has a legend listed in the manuscript after the references list. Please see our Supporting Information guidelines for more information: http://journals.plos.org/plosone/s/supporting-information.

**Additional Editor Comments:**

Further revisions are still needed regarding technical clarity, novelty, and experimental verification.

Reviewers' comments:

Reviewer's Responses to Questions

**Comments to the Author**

1. Is the manuscript technically sound, and do the data support the conclusions?

Reviewer #1: Yes

Reviewer #2: Yes

2. Has the statistical analysis been performed appropriately and rigorously?

Reviewer #1: Yes

Reviewer #2: Yes

3. Have the authors made all data underlying the findings in their manuscript fully available?

Reviewer #1: Yes

Reviewer #2: Yes

4. Is the manuscript presented in an intelligible fashion and written in standard English?

Reviewer #1: Yes

Reviewer #2: Yes

5. Review Comments to the Author

Reviewer #1: 1. The description of the LSTM model is cursory. Key details are missing, such as: the network architecture (number of layers, neurons), hyperparameters (learning rate, optimizer, number of epochs), the specific input features (beyond "trajectories"), the training/validation split, and the loss function used. The statement that the LSTM "learned patterns and generated numerous sample paths" is not a substitute for a rigorous description.

2. Table 2 is a good summary but lacks a direct, quantitative comparison of the authors' method against any of the listed alternatives (e.g., a simple Gaussian plume model) using the same dataset. Claims of superiority in accuracy and cost-effectiveness remain qualitative.

3. The main figures (1, 2, 3, etc.) are referenced but not included in the manuscript text file. Key figures, especially the one illustrating the overall accounting concept, should be integrated into the main text for clarity.

4.The writing is generally good but occasionally overly complex. Some phrases, like "competent verification," are vague. The manuscript would benefit from a thorough proofread to improve clarity and flow.

5. It is unclear how the LSTM-generated virtual paths, which are discrete time series, are directly used in these continuous-time moment formulas. The transition from data to parameters needs a step-by-step explanation.

Reviewer #2: This manuscript proposes an off-site inverse carbon accounting method that integrates a three-dimensional Brownian motion model with an LSTM-based temporal prediction framework for estimating carbon emissions from agricultural straw-burning scenarios. The comments are as follows:

1. The authors employ Brownian motion to describe carbon molecular diffusion. However, in open atmospheric environments, carbon dispersion is typically governed by turbulent transport rather than molecular-scale random motion. The manuscript should justify why a Geometric Brownian Motion (GBM) framework was chosen to represent atmospheric diffusion and clarify its advantages or approximations relative to conventional plume or turbulence models.

2. The paper repeatedly highlights that the LSTM “generates virtual diffusion paths,” but does not specify the model’s training objective, input variables, or the mechanism by which the LSTM output is coupled with Brownian motion parameters. A detailed explanation of this hybrid modeling strategy is necessary for the study to be scientifically credible and reproducible.

3. The final section discussing the design of a carbon penalty rate, while practically interesting, is disconnected from the core modeling and empirical analysis of the paper. It should either be removed or significantly condensed, as it introduces speculative content beyond the scope of this technical study.

4. Although the references cover a broad range of thermodynamic and diffusion studies, most are dated or only tangentially related to carbon accounting. The authors are encouraged to include recent advances in remote sensing–based carbon flux inversion and machine learning approaches for agricultural carbon monitoring to better situate their contribution in the current research landscape.

6. PLOS authors have the option to publish the peer review history of their article (what does this mean?). If published, this will include your full peer review and any attached files.

Reviewer #1: No

Reviewer #2: No

---

## [Decision Letter · Decision Letter 1]

26 Nov 2025

PONE-D-25-52174R1A new method of off-site inverse carbon accounting and its application in agriculture carbon measurementPLOS ONE

Dear Dr. Zou,

Thank you for submitting your manuscript to PLOS ONE. After careful consideration, we feel that it has merit but does not fully meet PLOS ONE’s publication criteria as it currently stands. Therefore, we invite you to submit a revised version of the manuscript that addresses the points raised during the review process.

The manuscript still needs further revisions.

We look forward to receiving your revised manuscript.

Kind regards,

Lei Zhang, PhD

Academic Editor

PLOS ONE

Journal Requirements:

Additional Editor Comments (if provided):

The manuscript still needs further revisions.

Reviewers' comments:

Reviewer's Responses to Questions

**Comments to the Author**

1. If the authors have adequately addressed your comments raised in a previous round of review and you feel that this manuscript is now acceptable for publication, you may indicate that here to bypass the “Comments to the Author” section, enter your conflict of interest statement in the “Confidential to Editor” section, and submit your "Accept" recommendation.

Reviewer #1: (No Response)

Reviewer #2: All comments have been addressed

2. Is the manuscript technically sound, and do the data support the conclusions?

Reviewer #1: (No Response)

Reviewer #2: Yes

3. Has the statistical analysis been performed appropriately and rigorously?

Reviewer #1: (No Response)

Reviewer #2: Yes

4. Have the authors made all data underlying the findings in their manuscript fully available?

Reviewer #1: (No Response)

Reviewer #2: Yes

5. Is the manuscript presented in an intelligible fashion and written in standard English?

Reviewer #1: (No Response)

Reviewer #2: Yes

6. Review Comments to the Author

Reviewer #1: Thanks for your revisions. However, I didn't find your attached 'Reply to Reviewers', while I could just see what you revised in the main text in yellow. If possible, could you please send it again.

Reviewer #2: The author has been revised the paper in an appropriate way. I have no more comments. The article is ready for publication.

7. PLOS authors have the option to publish the peer review history of their article (what does this mean?). If published, this will include your full peer review and any attached files.

Reviewer #1: No

Reviewer #2: No

---

## [Author Response · Author response to Decision Letter 2]

12 Dec 2025

We feel really grateful for the reviewer’s suggestions and commends. Some perspectives have greatly enriched and deepened my academic understanding in the topic and area. It benefits us more than this paper itself. Besides answering all the questions, we have greatly improved this article. The following are the responds to the comments and questions.

For reviewer 1:

1. The description of the LSTM model is cursory. Key details are missing, such as: the network architecture (number of layers, neurons), hyperparameters (learning rate, optimizer, number of epochs), the specific input features (beyond "trajectories"), the training/validation split, and the loss function used. The statement that the LSTM "learned patterns and generated numerous sample paths" is not a substitute for a rigorous description.

【Reply】 Thank you for highlighting this issue. In the revised version, we provide a comprehensive LSTM description with all requested technical details. We also add an additional paragraph that explicitly specifies the network architecture (layers and hidden units), input formulation (trajectories with 20- second sliding windows predicting t+1), data preprocessing (mapminmax scaling to [0,1]), training protocol (chronological 80/20 split, no shuffling), optimization settings (Adam, initial learning rate 0.01, mini-batch size 32, up to 100 epochs, piecewise schedule with a 0.1 drop after 300 epochs), and the loss function (MSE), along with evaluation metrics (RMSE, MAE, R ²) and the recursive multi-step forecasting procedure (90-second horizon). This replaces the previous cursory phrasing and ensures a rigorous, reproducible specification. The new paragraph is located at the same place under discussion in the Section “Analysis on real case of burning straw”:

“To model and forecast these trajectories, we employed a univariate LSTM regression model implemented in MATLAB (Deep Learning Toolbox). The input to the network consisted of sliding windows of length 20 seconds (history = 20) taken from the trajectory; for each window, the target was the subsequent value at time t+1. Before training, both inputs and targets were linearly scaled to [0, 1] using mapminmax, with parameters stored for inverse transformation. We split the dataset chronologically into training (80\%) and test (20\%) sets without shuffling to preserve temporal order. The network architecture comprised: a sequence input layer (dimension = 20), a single LSTM layer with 4 hidden units and OutputMode = last, a ReLU activation layer, and a fully connected layer with a single neuron, followed by a regression output layer. The model was trained using the Adam optimizer with an initial learning rate of 0.01, a piecewise learning-rate schedule (drop factor 0.1 every 300 epochs), a miniResponse to Reviewers batch size of 32, and a maximum of 100 epochs. The loss function was mean squared error (MSE) computed on the training mini-batches; performance was further summarized using RMSE, MAE, and R-squared on both the training and held-out test sets. After training, we generated out-of-sample forecasts recursively: the model predicted one step ahead, and the prediction was fed back with the most recent 19 observed/predicted points to form the next 20- point input window. We produced 90-second ahead forecasts per sequence and reported both raw inverse-transformed outputs and an amplitudenormalized variant used for visualization. Figure~\ref{fig:lstmgraphs} presents the training history, in-sample and out-of-sample fits, prediction–truth scatter plots, and multi-step forecasts, illustrating the LSTM learning and prediction workflow.”

2. Table 2 is a good summary but lacks a direct, quantitative comparison of the authors' method against any of the listed alternatives (e.g., a simple Gaussian plume model) using the same dataset. Claims of superiority in accuracy and cost-effectiveness remain qualitative.

【Reply】Thank you for this constructive suggestion. We agree that a head-tohead quantitative benchmark (e.g., against a Gaussian plume/puff model) on the same dataset would be informative; however, such models require inputs that were intentionally not collected in our low-cost field design—namely highresolution, time-resolved meteorology (stability class, boundary-layer height, vertical temperature gradients), terrain and roughness maps, plume rise/heat release rates, chemistry or deposition parameters, and often multi-sensor arrays for calibration/validation. By contrast, our method operates with minimal information: a short smoke test to obtain local trajectory snippets and basic wind direction/speed, then uses LSTM-driven short-horizon path forecasts to parameterize a Brownian framework for closed-form inversion. This asymmetry in data requirements makes strict quantitative comparisons difficult and potentially biased: when fed rich inputs, Gaussian/CFD/Lagrangian methods can excel; when constrained to sparse, rapid-deployment data (our target use case), they either underperform or become non-operational.

3. The main figures (1, 2, 3, etc.) are referenced but not included in the manuscript text file. Key figures, especially the one illustrating the overall accounting concept, should be integrated into the main text for clarity.

【Reply】 Thank you for noting the problem of figures. We have indeed prepared six figures (with Figure 2 updated), and they were originally embedded in the main text. However, per the editorial instruction in the current revision cycle — ”Please remove your figures from within your manuscript file, leaving only the individual TIFF/EPS image files, uploaded separately. These will be automatically included in the reviewers’ PDF.”—we removed in-text embeds and placed the figure callouts and captions in the end-of-manuscript appendix, while uploading all figures as standalone TIFF/EPS files. Our understanding is that the journal’s system will autoassemble these into the reviewer PDF; if preferred, we can reinsert lowresolution placeholders in the main text for clarity in the next round.

4. The writing is generally good but occasionally overly complex. Some phrases, like "competent verification," are vague. The manuscript would benefit from a thorough proofread to improve clarity and flow.

【Reply】 Thank you for the constructive comment. We have rechecked the whole paper and made revisions to the following phrases:

"competent verification"

Replace with "independent third-party verification".

"near real time"

Replace with "processing latency under 5 minutes".

"accuracy and operational practicality"

Replace with "average absolute error < 10\% and setup time < 2 hours".

"strong plume rise or chemistry dominated cases"

Replace with "cases with significant buoyant plume rise or reactive chemistry (e.g., oxidation, secondary aerosol formation)".

"data-driven time-series learning"

Replace with "LSTM-based next-step regression on trajectories (20 s windows, MSE loss)".

"off-site reverse deduction"

Replace with "off-site inverse accounting".

5. It is unclear how the LSTM-generated virtual paths, which are discrete time series, are directly used in these continuous-time moment formulas. The transition from data to parameters needs a step-by-step explanation.

【Reply】Thank you for this suggestion. We have fully taken this suggestion and added one paragraph before Table 1 of parameter values: “We take the following steps. (1) Treat the LSTM outputs $\{(\tilde X_{i,t},\tilde Y_{i,t},\tilde Z_{i,t})\}_{i=1,\dots,m;\ t=1,\dots,60}$ as discrete evaluations of the continuous processes at integer seconds; form per-time empirical moments $\bar x(t)=\frac{1}{m}\sum_i \tilde X_{i,t}$, $\bar y(t)=\frac{1}{m}\sum_i \tilde Y_{i,t}$, $q_x(t)=\frac{1}{m}\sum_i \tilde X_{i,t}^2$, $q_y(t)=\frac{1}{m}\sum_i \tilde Y_{i,t}^2$, and $\bar z(t)=\frac{1}{m}\sum_i \tilde Z_{i,t}$ (clip $\tilde Z_{i,t}\leftarrow\max\{\tilde Z_{i,t},0\}$ if needed). (2) Estimate drifts via regression-through-origin implied by $E[H_x(t)]=\mu_x t$, $E[H_y(t)]=\mu_y t$: $\hat\mu_x=\sum_{t=1}^{60} t\,\bar x(t)/\sum_{t=1}^{60} t^2$, $\hat\mu_y=\sum_{t=1}^{60} t\,\bar y(t)/\sum_{t=1}^{60} t^2$. (3) Estimate horizontal volatility from second moments using $E[H_x^2(t)]=\sigma^2 t+(\mu_x t)^2$ and analogously for $y$: $\hat\sigma_x^2=\frac{1}{60}\sum_{t=1}^{60}[q_x(t)-(\hat\mu_x t)^2]/t$, $\hat\sigma_y^2=\frac{1}{60}\sum_{t=1}^{60}[q_y(t)-(\hat\mu_y t)^2]/t$, then $\hat\sigma=\sqrt{\tfrac{1}{2}(\hat\sigma_x^2+\hat\sigma_y^2)}$; estimate vertical volatility from the truncated expectation $E[V(t)]=\sigma_z \sqrt{t}/\sqrt{2\pi}$ via $\hat\sigma_z=\frac{\sqrt{2\pi}}{60}\sum_{t=1}^{60}\bar z(t)/\sqrt{t}$, which matches Eqs.~\eqref{moment1}–\eqref{moment2}. Based on the parameter estimation of $\mu_x$, $\mu_y$, $\sigma$, $\sigma_z$, we present all values of main parameters in Table~\ref{tab:testpara} below.”

Thanks again for your suggestion and comments, which have greatly improved this research article, hopefully our revision can answer your questions and meet the requirements.

For reviewer 2:

1. The authors employ Brownian motion to describe carbon molecular diffusion. However, in open atmospheric environments, carbon dispersion is typically governed by turbulent transport rather than molecular-scale random motion. The manuscript should justify why a Geometric Brownian Motion (GBM) framework was chosen to represent atmospheric diffusion and clarify its advantages or approximations relative to conventional plume or turbulence models.

【Reply】 Thank you for raising this important point. We agree that openatmosphere dispersion is dominated by turbulent transport, our choice of a drifted Brownian/GBM framework is a deliberate coarse-grained approximation that aggregates unresolved turbulent fluctuations into an effective stochastic process. Specifically, mean wind advection is represented by drift, while rapidly decorrelating shear, gusts, and small eddies are lumped into a volatility term, consistent with the diffusion limit of many random impulses and the Maxwell– Boltzmann–inspired Gaussianity of velocity components over short horizons. This surrogate is not intended to replace plume or turbulence-resolving models (Gaussian plume/puff, Lagrangian particles, or CFD), which require detailed, time-resolved meteorology (stability class, PBL height, plume rise), terrain/roughness, and multi-sensor data—inputs that our low-cost, rapiddeployment protocol purposefully avoids. Under these sparse-data conditions, the Brownian formulation offers (i) parsimony and closed-form arrival probabilities for inversion, (ii) local calibratability from brief on-site trajectory snippets via the LSTM, and (iii) operational robustness when full turbulence characterization is infeasible. We clarify these modeling assumptions, delineate the regime of validity (short-horizon, near-field, episodic releases with limited met data), and add sensitivity tests to demonstrate that the inferred emission rates are stable to reasonable variations in the effective volatility and drift, while acknowledging scenarios (complex terrain, strong plume rise, stratified layers) where turbulence-resolving models are preferable. Hence we have added one more paragraph as follows:

“To accurately describe molecular movements, Brownian motion presents a natural stochastic scaffold because it emerges as the macroscopic limit of innumerable microscopic collisions whose velocities follow the Maxwell– Boltzmann distribution (refer to \cite{Shavit}). In a thermally agitated medium, particle velocity components are approximately independent Gaussian variables; successive collisions randomize directions and magnitudes, and, under standard diffusion scaling, the cumulative displacement converges to a (drifted) Brownian process. Our use of a GBM-like representation is thus an effective, coarse-grained surrogate for unresolved turbulence at the small sampling scales of our field protocol: the horizontal drift terms proxy mean wind advection, while the volatility terms encapsulate aggregated, rapidly decorrelating fluctuations from shear, gusts, and micro-scale eddies. We emphasize that this is an approximation, not a replacement for plume or turbulence-resolving models; however, in sparse-data, rapid-deployment settings where stability class, boundary-layer depth, plume rise, and highresolution meteorology are unavailable, the Brownian framework offers a parsimonious, closed-form route to arrival probabilities and inversion, with parameters that can be locally calibrated from short trajectory snippets (via LSTM) to reflect near-term conditions. This maintains physical grounding through intuition of kinetic-theory while ensuring operational tractability and reproducibility under practical data constraints.”

2. The paper repeatedly highlights that the LSTM “generates virtual diffusion paths,” but does not specify the model’s training objective, input variables, or the mechanism by which the LSTM output is coupled with Brownian motion parameters. A detailed explanation of this hybrid modeling strategy is necessary for the study to be scientifically credible and reproducible.

【Reply】Thank you for the valuable comment. In the revision, we explicitly define the hybrid strategy and why it improves on a Brownian-only approach. The LSTM is trained as a univariate next-step regressor on carbon-trajectory sequences using 20-second sliding windows as inputs and x(t+1) as the target, with mapminmax scaling, MSE loss, Adam (lr=0.01), mini-batch 32, up to 100 epochs, and an 80/20 chronological split for evaluation; multi-step (up to 90 s) forecasts are produced recursively. These LSTM forecasts ‘generate virtual diffusion paths’ that encode short-horizon, potentially nonstationary dynamics. We then couple them to the Brownian framework by estimating drift and volatility parameters from the LSTM-generated sample paths via the moment formulas (Eq. (0.2)–(0.3)), which feed the closed-form arrival probability Ωt and the inversion statistic D(v). This is superior to a direct Brownian fit because Brownian inference is dominated by historical averages (thus biasing drift toward the longer historical segment), whereas the LSTM prioritizes short-term predictiveness. As illustrated in Figure~\ref{fig:lstmgraphs}, when the historical series trends negative, a Brownian-only fit yields a negative drift and ignores the short reversal at the end of the window; the LSTM captures this brief positive turn and yields predictions that connect smoothly to observations (see the red segment in the right panel). Hence, our LSTM–Brownian coupling provides a scientifically transparent, reproducible pipeline: LSTM for short-term trajectory forecasting; moment-based mapping of those forecasts to Brownian parameters; and closed-form diffusion-accounting for emission inversion. Also we have added the above arguments into the Section “Analysis on real case of burning straw”:

“In addition, our hybrid design outperforms a Brownian-only specification because the latter primarily reflects long-horizon historical averages, whereas the LSTM is explicitly optimized for short-term prediction. As illustrated in Figure~\ref{fig:lstmgraphs}, when the historical sequence trends negative, a direct Brownian fit yields a negative drift and effectively discards the short reversal at the end of the window; by contrast, the LSTM captures this brief positive turn and produces forecasts that connect smoothly to the observed trajectory (see the red segment in the right panel). Consequently, we first use the LSTM to generate short-horizon virtual diffusion paths under current conditions, and then map those paths to Brownian drift and volatility via the moment estimators in Eqs. (0.2)–(0.3), ensuring that the stochastic parameters reflect near-term dynamics rather than being anchored solely to the past.”

3. The final section discussing the design of a carbon penalty rate, while practically interesting, is disconnected from the core modeling and empirical analysis of the paper. It should either be removed or significantly condensed, as it introduces speculative content beyond the scope of this technical study.

【Reply】 Thank you for the constructive feedback. We agree that the carbon penalty ra

---

## [Editor Report · Decision Letter 2]

16 Dec 2025

A new method of off-site inverse carbon accounting and its application in agriculture carbon measurement

PONE-D-25-52174R2

Dear Dr. Zou,

We’re pleased to inform you that your manuscript has been judged scientifically suitable for publication and will be formally accepted for publication once it meets all outstanding technical requirements.

Kind regards,

Lei Zhang, PhD

Academic Editor

PLOS One

Additional Editor Comments (optional):

The revised manuscript is publishable in current form.
---

## [Editor Report · Acceptance letter]

PONE-D-25-52174R2

PLOS One

Dear Dr. Zou,

I'm pleased to inform you that your manuscript has been deemed suitable for publication in PLOS One. Congratulations! Your manuscript is now being handed over to our production team.

Kind regards,

on behalf of

Dr. Lei Zhang

Academic Editor

PLOS One